# A New Treatment Landscape for RCC: Association of the Human Microbiome with Improved Outcomes in RCC

**DOI:** 10.3390/cancers15030935

**Published:** 2023-02-01

**Authors:** Xuan-Mei Piao, Young Joon Byun, Chuang-Ming Zheng, Sun Jin Song, Ho Won Kang, Won Tae Kim, Seok Joong Yun

**Affiliations:** 1Department of Urology, College of Medicine, Chungbuk National University, Cheongju 28644, Republic of Korea; 2Department of Emergency, Chungbuk National University Hospital, Cheongju 28644, Republic of Korea; 3Department of Urology, Chungbuk National University Hospital, Cheongju 28644, Republic of Korea

**Keywords:** renal cell carcinoma, immune checkpoint inhibitor, microbiome

## Abstract

**Simple Summary:**

Renal cell carcinoma (RCC) is a highly vascularized and immunogenic tumor, and its treatment has been revolutionized by the development of immune-checkpoint inhibitors. However, the clinical benefit of these agents is limited in patients with metastatic disease. The microbiome has emerged a novel therapeutic target in recent years and has shown promising values. Understanding the microbiome of the tumor microenvironment is essential for the treatment of RCC.

**Abstract:**

Microbes play different roles in metabolism, local or systemic inflammation, and immunity, and the human microbiome in tumor microenvironment (TME) is important for modulating the response to immunotherapy in cancer patients. Renal cell carcinoma (RCC) is an immunogenic tumor, and immunotherapy is the backbone of its treatment. Correlations between the microbiome and responsiveness to immune checkpoint inhibitors have been reported. This review summarizes the recent therapeutic strategies for RCC and the effects of TME on the systemic therapy of RCC. The current understanding and advances in microbiome research and the relationship between the microbiome and the response to immunotherapy for RCC are also discussed. Improving our understanding of the role of the microbiome in RCC treatment will facilitate the development of microbiome targeting therapies to modify the tumor microbiome and improve treatment outcomes.

## 1. Introduction

### 1.1. Epidemiology and Classification of Renal Cell Carcinoma

Renal cell carcinoma (RCC) originates within the renal cortex and accounts for approximately 90% of primary malignant renal tumors. The incidence of RCC varies widely from region to region, with the highest incidence reported in Western countries [1]. The American Cancer Society estimated that approximately 79,000 new cases of kidney cancer (50,290 in men and 28,710 in women) would be diagnosed and approximately 13,920 people (8960 men and 4960 women) would die from this disease in the United States in 2022 [2]. Worldwide, more than 400,000 new cases of RCC and more than 170,000 deaths from kidney cancer occur annually [3].

RCC comprises different subtypes with specific histopathological and genetic characteristics. The 2016 World Health Organization (WHO) classification categorized RCCs into three main histological types [clear cell RCC (ccRCC) (70–80%), papillary RCC (10–15%), and chromophobe RCC (4–5%)]; however, RCCs comprise a broad spectrum of histological entities [4]. The prognosis differs among the different RCC types, which are closely related to the different treatment strategies for different subtypes. ccRCC is the most common histological subtype, and approximately 40% of ccRCC patients develop metastases, resulting in a 5-year survival rate of 10% [5]. This year, the WHO published a new 2022 classification that introduced a molecular-driven renal tumor classification [6]. This novel molecular classification includes SMARCB1-deficient medullary RCC, TFEB-altered RCC, Alk-rearranged RCC, and ELOC-mutated RCC. Eosinophilic solid and cystic RCC is a novel morphologically defined RCC entity. The integration of classic histologic diagnoses with advanced molecular techniques, such as methylation profiling, RNA sequencing, whole-genome sequencing, or whole-exome sequencing, is a prerequisite for the design of personalized therapeutic strategies. Although it is important to include a pathologist/molecular expert on the design team of future clinical trials, many institutions do not yet have the advanced molecular tools [6].

### 1.2. Treatment of Renal Cell Carcinoma

Complete surgical tumor resection is the standard of care for patients presenting with localized RCC [1,7]. However, approximately one-third of patients treated with curative intent will develop metastatic disease recurrence [8]. Earlier treatment for patients with metastatic disease [including the initially diagnosed metastatic RCC (mRCC) patients] relied on several cytokines, such as interferon alfa (IFN-α) and interleukin-2 (IL-2), which aimed to activate the antitumor immune system [9]. However, because of the deficiencies and severe side effects, systemic therapies including targeted therapy and immunotherapy, have been strongly recommended since 2006 [5]. Tyrosine kinase inhibitors (TKIs), including the mechanistic target of rapamycin (mTOR) inhibitors and a vascular endothelial growth factor (VEGF) antibody, are the most widely used targeted drugs [1] and they are considered as anti-angiogenic agents. Because RCC is a highly vascularized tumor, anti-angiogenic therapies achieve disease stabilization or regression and prolonged survival in up to 30% of mRCC patients. The suggested immunotherapeutic strategies, which are known as immune checkpoint inhibitors (ICIs), target and block the inhibitory T-cell receptor PD-1 or cytotoxic T-lymphocyte-associated antigen 4 (CTLA-4) to restore tumor-specific T-cell immunity [10]. PD-L1/L2, the ligands for the PD-1 receptor, and CD80/86, the ligands for CTLA-4 are also common targets of ICIs [11]. The International Metastatic RCC Database Consortium (IMDC) risk model is used to predict the survival of mRCC patients treated with systemic therapy [1,12], and the updated 2022 European Association of Urology (EAU) guidelines for mRCC treatment according to the IMDC risk recommend the concomitant use of TKIs and ICIs or two kinds of ICIs. The management of mRCC from the updated 2022 EAU guideline is shown in Figure 1.

Although advances in systemic therapies have improved the life expectancy of patients with advanced metastatic disease, there is still an unmet need to identify novel targeted therapies for mRCC patients [7]. The heterogeneous nature of RCC is an important challenge that needs to be overcome. An improved understanding of the pathophysiology of the disease has promoted research efforts to produce novel therapeutic agents targeting specific biological pathways [13]. Most published trials have focused on clear cell carcinoma subtypes, and there are no robust evidence-based recommendations for non-ccRCC subtypes, which is an issue that needs to be addressed.

## 2. The Tumor Microenvironment in Renal Cell Carcinoma Biology and Therapy

The tumor microenvironment (TME) is the ecosystem that surrounds a tumor, and it is composed of immune cells, the extracellular matrix, blood vessels, and fibroblasts; it provides the factors necessary for cancer growth, invasion, and angiogenesis [14,15]. Because the tumor and the TME interact and influence each other, effective cancer treatments rely on a combination of drugs that target multiple components of TME, such as the vascular system that feeds the tumor cells, the immune system that fights or helps the tumor cells, and the cancer cells themselves [16]. Thus, in the treatment of RCC, which is a highly immunogenic and pro-angiogenic cancer, it is critical to understand the TME because it affects the response to treatment. The aim of current treatments for mRCC is to block the nutrition suppliers, namely, “angiogenesis,” and to attack the enemies, the “tumor cells,” using a proper double-edged sword, which is the “immune system.” The interplay between the immune system and angiogenesis in RCC is described in Figure 2.

Angiogenesis is one of the hallmarks of cancer, and malignant tumors secrete multiple angiogenic growth factors simultaneously. These factors, which are involved in tumor growth and survival, are targeted by anti-angiogenesis treatments, mainly TKI therapies, including the anti-VEGF, anti-mTOR, and the recently introduced anti-MET [also tyrosine-protein kinase Met or hepatocyte growth factor receptor (HGFR)], anti-RET, anti-platelet derived growth factor receptor (PDGFR), and anti-fibroblast growth factor receptors (FGFRs); these therapies have become a weapon of fundamental importance [17]. In addition, angiogenesis promotes metastasis by inducing the formation of abnormal neovessels, which provide an escape route for the tumor cells to enter the circulation [18,19]. These abnormal vascular networks influence the TME and form a hostile microenvironment characterized by hypoxia and acidosis, which in turn promotes tumor angiogenesis and reduces the effect of anti-tumor treatments such as immunotherapy [19,20]. The hypoxic TME creates an inhospitable environment for immune cells; it induces immune tolerance and immune escape by interfering with the tumor-killing function of effector cells and preventing their homing to the TME; and it promotes immune-suppressive circumstances in part by increasing checkpoint molecules [21]. To prevent this vicious cycle in the TME of RCC, normalization of tumor vasculature by restoring the balance of pro-angiogenic and anti-angiogenic factors is an essential concept that may produce a hostile microenvironment and activate the immune system [19,20].

ccRCC is extensively infiltrated with leukocytes such as CD8+ T cells, CD4+ T cells, and natural killer (NK) cells, as well as myeloid cells including macrophages and neutrophils [22]. The immune function in ccRCC can be suppressed by the inhibitory effects of regulatory T (Treg) cells and myeloid cell types such as myeloid-derived suppressor cells (MDSCs), macrophages, and neutrophils in the TME [5]. Such a TME restricts the effectiveness of immune surveillance as well as the activity of ICIs. However, inhibition of antitumor immune responses mostly depends on the expression of key receptors, namely, “immune checkpoints” on the surface of T cells that prevent full T cell activation [22]. The most studied immune checkpoints are CTLA-4 and PD1, along with its ligand PD-L1 [22,23,24]. Few patients with RCC derive durable benefits from ICIs, underscoring the need to identify reliable biomarkers to predict the response to ICIs and to develop new therapeutic targets. Improving our knowledge of the functions of immune mediators within the TME in RCC could assist in developing novel therapies.

## 3. The Microbiome as a Novel Member of the Tumor Microenvironment in Renal Cell Carcinoma

Recently, unexpected guests, the microbiota have emerged as an important part of the TME in many types of cancer and are shown to affect tumorigenesis and tumor progression [25]. The TME is an attractive niche for microbial growth, and microorganisms in human tumors have been identified for over a century [26]; however, because of technological limitations, the breadth of microorganisms and the depth of their influence has only been moderately appreciated to date. The development of sequencing techniques has improved the study and manipulation of the intratumoral microbiome to enhance the clinical response to cancer treatments.

### 3.1. What Is the Microbiome?

Since 2007, the Human Microbiome Project (HMP) has been charged with the mission of facilitating the comprehensive characterization of the human microbiome and analyzing its role in the human health and disease [27]. The creation of the HMP accelerated the development of technologies for exploring the human microbiome and promoted the prosperity of this research field formally.

The human microbiome is the collective genomes and by-products of all microorganisms such as bacteria, viruses, and fungi that inhabit the human body [28]. These microbiotas are located throughout the human body but largely in the human mouth, skin, and gut to affect digestion, shape the immune system, and even influence one’s mood and behavior [29,30,31]. The microbiome in the human body is extensive, containing at least a 100-fold greater number of unique genes than the human host genome, and the distribution of commensal microbes across anatomical sites reveals distinct microbial communities [25,27]. These microbes form an evolutionary partnership with humans and influence most of the essential physiological functions, such as metabolism, tissue development, and host defense [32]. Accordingly, studies have identified correlations between the microbiome and metabolic disorders, cardiovascular diseases, neurological disorders, and even psychological disorders such as schizophrenia [33,34,35]. The role of the microbiome in cancer has been increasingly investigated in recent years.

### 3.2. How Does the Microbiome Affect Cancers?

The link between the tumor microbiome and cancer is mediated by four main mechanisms, as follows: (1) tumor microbiome-induced gene mutations directly promote tumorigenesis; (2) the tumor microbiome regulates oncogenes or oncogenic pathways; (3) the tumor microbiome modulates the host immune system; and (4) the tumor microbiome produces small molecules or metabolites that influence cancer development, progression, and response to therapeutic agents [25,28]. The origin of the tumor microbiome remains unclear. If the tumor microbiome is directly related to tumor formation, tumor microbiomes with latent oncogenic features may exist in the human body waiting for a stimulus to induce tumorigenesis. On the other hand, the TME could attract microbiota to accumulate after tumor formation through multiple mechanisms. Considering that immune escape is one of the features of cancer cells, microorganisms could avoid immune clearance under the “shelter” of the inadequate strength of immune cells inside the tumor. The hypoxic condition in tumors provides an ideal environment for the survival of anaerobic bacteria. The highly nutritious conditions inside tumor tissues also attract bacteria and provide metabolites such as ribose and aspartic acid. In addition, the highly disorganized neovascularization, slow blood flow, and blood leakage inside the tumor can facilitate the movement of bacteria from the blood circulation to the tumor tissue [36,37,38]. The interaction between the tumor microbiome and the TME not only promotes tumorigenesis and progression but also regulates the response to cancer immunotherapy.

### 3.3. The Microbiome in Renal Cell Carcinoma

The microbiome of RCC is the least studied among genitourinary cancers [39,40]. In 2004, a population-based analysis by the Iowa Cancer Registry found that a history of urinary tract infection of the bladder or kidney is associated with an increased risk of developing RCC, particularly in men who smoke, suggesting the presence of complex interactions between bacteria and epidemiological risk factors in RCC [41]. In 2019, the first attempt at characterizing the microbiome in renal tissue was achieved by Stefan et al. using 10 formalin-fixed paraffin-embedded tissue samples from five RCC patients with no history of urinary tract infections in the last 6 months [42]. The study revealed that both healthy kidneys and RCC tissues contained a plethora of specific microorganisms, and the microbiome differed significantly between benign and malignant tissues. However, because of the small sample size, further validation is necessary for a large cohort. More recently, a study from China analyzed the composition of the gut microbiota in 51 ccRCC patients and 40 healthy controls using 16S rRNA sequencing. The results showed that *Blautia*, *Streptococcus*, [*Ruminococcus*]_torques_group, *Romboutsia*, and [*Eubacterium*]_hallii_group are dominant and positively associated with ccRCC [38]. In particular, *Streptococcus lutetiensis* promotes ccRCC proliferation, migration, and invasion in-vitro via the TGF-signaling pathway, which could be a target for RCC treatment. The main limitation of this study was the 16S rRNA sequencing method used, which can only identify the microbiota composition at the genus level. Thus, to identify the exact strains, a more precise detection method such as metagenomic sequencing can be applied in future evaluation. The same genus, *Streptococcus*, (but a different species, the *epidermidis*), was more abundant in primary ccRCC tumors than in the thrombus or normal adjacent tissues in another study [43]. In contrast to these studies, which indicated that the diversity of the microbiome is increased in RCC, a study showed that species diversity is decreased in RCC tissues [44]. The study demonstrated that 25 taxa increased and 47 decreased in RCC tissues compared with normal tissues; among them, the class Chloroplast and the order Streptophyta showed a considerable ability to discriminate RCC tissues from normal tissues. These RCC studies focused on ccRCC, which is the largest group, and other RCC subtypes need to be included in the analysis. A comparison of the three main subtypes of RCC (ccRCC, pRCC, and chRCC) and normal controls was recently performed to identify the specific microbiome in each group and to detect new prognostic markers of RCC using tumor microbiome and stromal inflammatory markers [45]. The study showed that the alpha diversity was significantly altered when comparing the normal kidney with all types of RCC tissues, and the bacterial burden was higher in adjacent normal tissues than that in tumor tissues. It is worth noting that this study offered the first theory that there is a noteworthy correlation between bacterial burden and the content of PU.1+ macrophages and CD66b+ neutrophils in RCC. Specifically, tumors with a high content of PU.1+ and CD66b+ cells in the stroma have a lower bacterial burden, and in those tumors with a high bacteria burden, patients with a relatively increased number of PU.1+ cells and CD66b+ cells suffered poor prognoses. The associations forecast the potential of the tumor microbiome in combination with the properties of stromal cells as prognostic markers.

## 4. The Microbiome and Immunotherapy of Renal Cell Carcinoma

Earlier detection of small renal masses is associated with a significant stage migration of RCC at presentation, and the development of novel therapies increases the life expectancy of RCC patients. Except for the potential contribution of the microbiome to the etiology of RCC, studies have provided information on the responsiveness of RCC to certain drugs such as ICIs and TKIs. Resistance to ICIs is associated with impaired immune modulation caused by intrinsic or extrinsic factors including complex components of TME and their associations in the TME. Tumor cell-intrinsic factors that contribute to immunotherapy resistance include the expression or repression of certain genes and pathways in tumor cells, such as constitutive PD-L1 expression, loss of tumor antigen expression, and alterations in the antigen-presenting and processing machinery that prevent immune cell infiltration or function within the TME. These mechanisms may exist at the time of initial presentation, which constitutes primary resistance, or evolves later after an initial response, thus forming part of adaptive resistance mechanisms. The tumor cell-extrinsic factors include inhibitory immune checkpoints (CTLA-4, PD1, and others), T cell exhaustion and phenotype change, immune suppressive cell populations (Tregs, MDSCs, and type II macrophages), and the release of cytokines and metabolites in the TME [46].

### 4.1. The Microbiome Related to the Response to ICI

Recent studies identified a specific microbiome in patients who are sensitive to ICIs. Of five studies that used different sequencing methods, four identified the same bacterium, *Akkermansia muciniphila*, that was associated with the response to ICIs [47,48,49,50]. This may be related to an immune regulatory function of *A. muciniphila*, resulting in an increase in CXCR3^+^CCR9^+^CD4^+^ T cells and the upregulation of IL-12 and the function of dendritic cells (DCs) to improve the effect of PD-1 inhibitors [47,51]. This immunomodulatory effect may be related to the production of metabolites, mainly short-chain fatty acids (SCFAs) released by *A. muciniphila*. SCFAs regulate tumorigenesis either by suppressing the activity of histone deacetylase to inhibit transcription factors involved in tumorigenesis or indirectly by modulating inflammation [52]. *Alistipes putredinis*, also discovered by Routy et al., modulates the host immune system by recruiting unique memory CD8^+^ T cells and NK cells to the periphery to enhance the PD-1 blockade effect [47]. *Bacteroides*, *Bifidobacterium*, *Firmicutes*, and *Faecalibacterium* sp. are also associated with the response to ICIs in RCC patients [48]. Bacteroides species induce adaptive T-cell-mediated immune responses by producing capsular polysaccharides [53]. *Bifidobacterium* also induces immune responses by increasing tumor-infiltrating lymphocytes, promoting the maturation of DCs, up-regulating the expression of IFN-γ and pro-inflammatory cytokines, and priming tumor-specific CD8^+^ T cells [54,55]. A high level of CD8^+^ T cells is associated with longer survival in melanoma patients [56], suggesting that RCC patients with increased *Bifidobacterium* are more sensitive to ICIs. *Faecalibacterium* sp. induces the proliferation of CD4^+^ or CD8^+^ T cells, increases the production and differentiation of Treg cells, and upregulates the expression of inducible T cell costimulatory, thereby decreasing the risk of ICI-related colitis and improving the response to ICIs in melanoma patients [57]. Along with *Firmicutes*, these microbiome components improve the clinical outcome of patients treated with ICIs by producing SCFAs and regulating the immune response. In patients with mRCC, *Stenotrophomonas maltophilia* and *Corynebacterium* sp. are increased in ICI responders [58]. These data led to the design of a phase I randomized, prospective, open-label clinical trial (NCT05122546) in 2021 to investigate the efficacy of oral administration of live *Clostridium butyricum* MIYAIRI 588 (CBM 588) in combination with first-line ICIs (ipilimumab and nivolumab combo) in patients with intermediate or poor-risk mRCC, and to evaluate whether this could modulate the gut microbiome of these patients [59]. CBM588 is a bacterial strain that restores *Bifidobacterium* sp. in the microbiome. The initial results of this clinical trial showed that patients who received ICIs plus CBM-588 had better progression-free survival (PFS) and overall survival (OS) than those treated with ICIs alone. Analysis of the microbiome identified several increased (*Bifidobacterium* sp., *Bifidobacterium longum*, and *Butyricimonas faecalis*) and decreased (*Desulfovibrio* sp.) microorganisms in responders receiving CBM-588 [59]. No differences in toxicities were observed. This clinical trial is ongoing, with an estimated completion date in November 2023 [60]. The identified microbiome associated with ICI responses and the possible mechanisms are depicted in Figure 3.

### 4.2. Effects of Environmental Factors That Alter the Gut Microbiome on ICIs

Several factors that alter the composition of the gut microbiome affect the interaction of the microbiome with the immune system, thus determining the hosts’ response to treatment. Environmental factors such as the administration of antibiotics or proton pump inhibitors (PPIs) are the most important factors affecting the response to ICIs. A study that included 603 RCC patients treated with ICIs from ten study groups showed that antibiotics decrease PFS and OS rates [47,49,61,62,63,64,65,66,67,68,69,70]. The study indicated that there is a controversial link between PPIs and anticancer agents including ICIs, which may be due to the dysbiosis of the gut microbiota; however, the link is not strong [71,72]. A retrospective study that examined the impact of antibiotics and PPIs on the efficacy of and tolerance to ICIs in different cancers including RCC showed that the use of antibiotics and PPIs alone or in combination has a negative impact on PFS and OS [73]. However, a different study showed no association between the concomitant use of PPIs and ICIs and the survival outcomes of patients with mRCC [74]. The effect of TKIs combined with antibiotics targeting *Bacteroides* species was evaluated by Hahn et al., and the results showed that PFS was increased in RCC patients receiving first-line VEGF-TKIs with antibiotics [75]. However, further research is warranted, because the exact mechanisms by which the immune checkpoint inhibitors and antibiotics affect each other remain unclear.

A summary of microbiome studies on RCC presented in Table 1.

### 4.3. Fecal Microbiota Transplantation for RCC Treatment

Maintenance of gut homeostasis is essential for human health due to the bidirectional feedback loop that exists between the host-associated microbiota and human health [76]. Many human diseases including cancer associated with the dysbiosis of gut microbiota [77,78,79]. Homeostasis and dysbiosis are not as simple as the presence or absence of specific microbial species because the microbial communities show high diversity between individuals and the complex interactions exist within a community of microbes, all of which can have a marked impact on host health and disease and are difficult to replicate artificially with therapeutics [80]. Several strategies, including dietary interventions, the use of probiotics and antibiotics, and fecal microbiota transplantation (FMT) have been used to shape the gut microbiota composition during homeostasis and aid in cancer treatment [81]. Anti-cancer effect of FMT is achieved by transplanting the feces from healthy donors to diseased recipients to restore homeostasis with beneficial bacteria, which can have a positive impact on the recipient’s immune system and response to treatment [81]. The complex interactions among a whole community of microbes and the balance between metabolites and immune cells, which is difficult to achieve with the administration of single metabolites or microbes can be restored by FMT from healthy donors. The beneficial effects of this approach made it a medical breakthrough in recent years [82]. In vivo studies revealed that ICI-responsiveness could be acquired by transplanting feces from ICI-responsive RCC patients into germ-free mice [61,83]. This response was found to be diminished with the administration of antibiotics [61]; however, the decrease was reversible, and the non-responding mice could be rescued by FMT from responding donors, as well as by oral administration of immunostimulatory microbes [83]. The role of FMT in the systematic treatment of RCC remains to be investigated scientifically and clinically. A randomized clinical trial showed that FMT greatly relieves diarrhea induced by TKIs in mRCC patients and indicated that *A. muciniphila* is commonly present in abundance in both healthy donors and patients with improved outcomes [84]. Current clinical trials are investigating the effect of FMT on patients with RCC receiving ICIs and evaluating its role in the improvement of treating or preventing immune-related toxicities (NCT04163289; NCT04758507) [80]. However, the use of FMT in clinical practice is challenging because of the stigma associated with providing or accepting stool samples [46].

## 5. Immunotherapy in Non-Clear Cell Renal Cell Carcinoma

Other kidney cancer subtypes, such as pRCC and chRCC, differ from ccRCC in the prognosis and treatment strategies. These RCCs are included in the non-clear cell RCC (nccRCC) group, and each type has unique molecular drivers that differ from those of ccRCC [85]. For nccRCC patients, targeted treatments that are commonly used for ccRCC, including ICIs, have not significantly improved patient survival [86]. The heterogeneous molecular background of nccRCC needs to be considered in the design of personalized targeted strategies. Although recent clinical trials are selecting patients based on specific disease histology and conducting fewer all-inclusive basket trials, new therapies are still limited in the clinical practice [87]. A broad research approach is necessary to gain insight into alterations other than genetic factors, such as the connections within the TME and the role of the tumor microbiome, which will benefit patients diagnosed with these rare cancers.

## 6. Conclusions

The clinical use of ICIs in mccRCC has limited benefits on the survival outcomes of patients, and many patients experience resistance to ICIs. The response rates to ICIs are even lower in other types of RCC. Predicting the response to ICIs and designing strategies to overcome ICI resistance are unmet needs to improve the survival of RCC patients. The dynamic interplay between the TME and treatment response has been a subject of interest in recent years. The microbiome was identified as a novel component of the TME, and the association of the microbiome with the response to anticancer therapy, as well as with the incidence of adverse events, has been reported by several studies. This review discusses the systemic therapies used to treat mRCC and the intimate relationship between the microbiome and the response to immunotherapy for RCC. The microbiome participates in host immune responses and regulates antigen presentation and T-cell initiation and activation. Elucidating the immune mechanism of the microbiome and the dynamic interactions between the microbiome and other factors in the TME is not only essential to predict the prognosis of patients but also for the development of microbiota-based anticancer immunotherapies for the treatment of different RCC histological types. Ultimately, the objective of prospective studies is to validate novel biomarkers that can be used and integrated into clinical practice. Furthermore, innovative biotechnological strategies are needed for the clinical application of the microbiome as an anti-cancer treatment target. Over the past few decades, the successful application of nanotechnology in cancer diagnosis and treatment has made it possible to target specific oncogenic microbes by intervening with the microbiome. This is an instant hit and brings the nascent field of microbiome intervention in cancer, however, it still has many aspects that need to be improved.

## Figures and Tables

**Figure 1 cancers-15-00935-f001:**
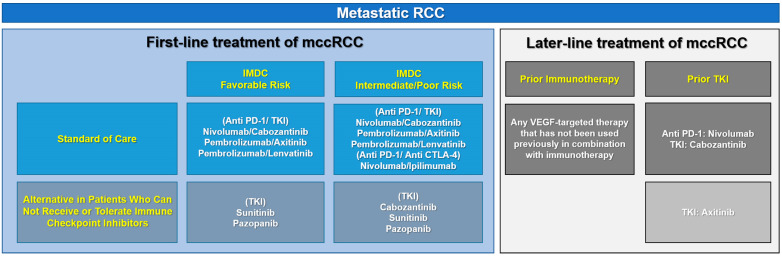
Management of mccRCC from the updated 2022 EAU guideline. Systemic treatment with a combination of anti-angiogenetic agents (mainly TKIs) and immune checkpoint inhibitors (mainly anti-PD-1/PD-L1/CTLA-4) is recommended for mccRCC patients. The combinations vary according to the IMDC risk stratification of RCC. mccRCC, metastatic clear cell renal cell carcinoma; EAU, European Association of Urology; IMDC, International Metastatic RCC Database Consortium; TKI, tyrosine kinase inhibitor; VEGF, vascular endothelial growth factor.

**Figure 2 cancers-15-00935-f002:**
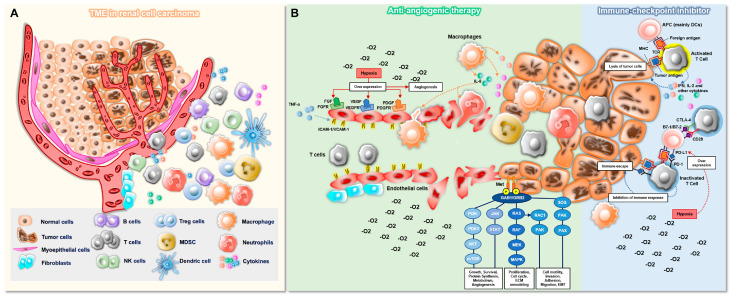
Interaction between angiogenesis and the immune system in renal cell carcinoma. (**A**) Components of the TME in renal cell carcinoma. (**B**) Role of TME factors in regulating angiogenesis and the immune system. The systemic targeting therapies according to their roles include anti-angiogenic agents targeting FGFR, VEGFR, PDGFR, mTOR, and Met and immune checkpoint inhibitors targeting PD-1, PD-L1, and CTLA-4, which are mainly used in clinical practice. TME, tumor microenvironment; ICAM-1, intercellular adhesion molecule 1; VCAM-1, vascular cell adhesion molecule 1; FGF and FGFR, fibroblast growth factor and fibroblast growth factor receptor; VEGF and VEGFR, vascular endothelial growth factor and vascular endothelial growth factor receptor; PDGF and PDGFR, platelet-derived growth factor and platelet-derived growth factor receptor; MDSC, myeloid-derived suppressor cell; NK cell, natural killer cell; Treg cell, regulatory T cell; APC, antigen-presenting cell; MHC, major histocompatibility complex; TCR, T cell receptor; IL, interleukin; IFN, interferon.

**Figure 3 cancers-15-00935-f003:**
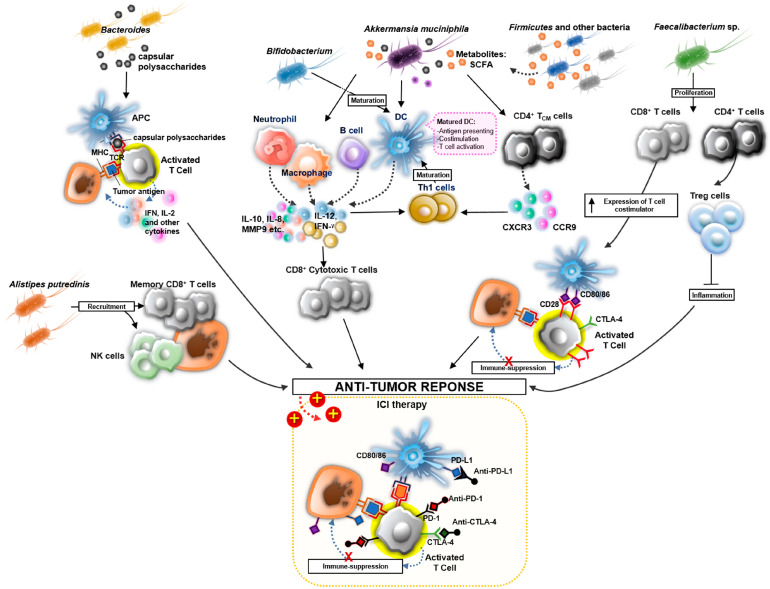
Microbiome related to the response to ICI. Several microbiomes were identified to improve the responses to ICIs directly by stimulating primary and adaptive immune responses to kill tumor cells and indirectly by inducing the release of SCFAs to activate immune responses. The microbiome induced the overexpression of T cell costimulatory, which activates T cells involved in anti-tumor responses. SCFA, short chain fatty acid; NK cell, natural killer cell; Treg cell, regulatory T cell; APC, antigen-presenting cell; MHC, major histocompatibility complex; TCR, T cell receptor; IL, interleukin; IFN, interferon; MMP9, matrix metalloproteinases 9; CXCR3, C-X-C motif chemokine receptor 3; CCR9, C-C motif chemokine receptor 9. ↑ represents the increased expression of T cell costimulatory.

**Table 1 cancers-15-00935-t001:** Comprehensive summary of microbiome studies in RCC.

Different Microbiome in Normal and RCC Tissues.						
Study	Year of Publication	Number of Specimens	Method	Microbiome Diversity in RCC	Specific Microbiome in RCC	Specific Microbiome in Normal Control
Heidler et al. [42]	*2019*	Ten FFPE tissue samples (malignant tissues and tumor-free renal cortex tissues) of five RCC patients	16S rRNA sequencing	Increased	*Cyanophora paradoxa* *Spirosoma Navajo* *Phaeocystis antarctica* *Euglena mutabilis* *Mycoplasma vulturii*	*Microbacterium* *Pelomonas* *Staphylococcus* *Strepotococcus* *Leuconostoc garlicum* *Corynebacterium vitaeruminis* *Anaerococcus nagyae* *Ethanoligenens harbinense* *Neisseria bacilliformis* *Thermicanus aegyptius* *L. mesenteroides*
					Found in both normal and cancer tissues, but more frequent in cancer tissues: *Aeromonas salmonicida* *Pseudoalteromonas haloplanktis* *Parageobacillus toebii* *Trachelomonas volvocinopsis* *M. mycoides* *Halomicrobium mukohataei*	
Chen et al. [38]	*2022*	Fecal samples from 51 ccRCC and 40 healthy controls	16s rRNA sequencing	Increased	*Blautia* *Streptococcus* *[Ruminococcus]_torques_group* *Romboutsia* *[Eubacterium]_hallii_group*	*Prevotella* *Lachnospira* *Lachnoclostridium* *Roseburia*
Liss et al. [43]	*2020*	Eighteen fresh frozen tissue samples (normal adjacent renal parenchyma, tumor, and thrombus tissues) of six RCC patients	Illumina HiSeq 3000	Increased	More abundant in tumor specimens than in normal adjacent kidney and tumor thrombus: *Micrococcus luteus* *Fusobacterium nucleatum* *Streptococcus agalactieae* *Corynebacterium diphtheriae*	
Kovaleva et al. [45]	*2022*	Forty FFPE tissue samples (10 ccRCC, 10 pRCC, and 10 chRCC; 10 normal kidney tissues)	16s rRNA sequencing	Phylum level: No differences	*Tenericutes phylum in ccRCC, pRCC*	*Gemmatimonadetes* *Chloroflexi* *Fusobacteria* *Parcubacteria* *Verrucomicrobia phyla*
				Genus level: Decreased	*ccRCC:* *Cutibacterium* *Sphingomonas* *Roseomonas* *Staphylococcus* *Mesomycoplasma* *Massilia* *Escherichia_Shigella* *Photobacterium* *pRCC:* *Cutibacterium* *Corynebacterium* *Escherichia_Shigella* *Clavibacter* *Enhydrobacter* *Phyllobacterium* *Mesomycoplasma* *Simplicispira,* *chRCC:* *Escherichia_Shigella* *Novosphingobium* *Cutibacterium* *Psychrobacter* *Lactococcus* *Acinetobacter* *Jeotgalicoccus* *Corynebacterium*	*Kocuria* *Phyllobacterium* *Micrococcus* *Cutibacteriu* *Corynebacterium* *Rothia* *Streptococcus* *Acinetobacter*
Wang et al. [44]	*2021*	Forty-eight tissue samples (malignant and normal adjacent tissues) of 24 RCC patients	16s rRNA sequencing	Decreased	*Phylum:* *Chlorofexi* *Classes:* *Nitriliruptoria,* *Nostocophycideae* *Orders:* *Deinococcales,* *Actinomycetales,* *Nitriliruptorales, Nostocales,* *Oceanospirillales* *Families:* *Deinococcaceae,* *Actinomycetaceae,* *Gordoniaceae,* *Pseudonocardiaceae,* *Nitriliruptoraceae,* *Nostocaceae,* *Acetobacteraceae* *Genera:* *Nitriliruptor,* *Deinococcus,* *Actinomyces, Gordonia,* *Pseudoclavibacter,* *Microlunatus,* *Amycolatopsis,* *Weissella,* *Brevundimonas, Phyllobacterium*	*Phylum:* *Cyanobacteria* *Classes:* *Coriobacteriia,* *Anaerolineae,* *Chloroplast,* *Erysipelotrichi, Gemmatimonadetes,* *Pedosphaerae* *Orders:* *Bifidobacteriales, Coriobacteriales, Caldilineales,* *H39,* *SJA_15,* *Streptophyta, Erysipelotrichales, Gemmatimonadales,* *Rickettsiales, Burkholderiales, Enterobacteriales, Pedosphaerales* *Families:* *Cellulomonadaceae,* *Bifdobacteriaceae, Coriobacteriaceae, Marinilabiaceae, Caldilineaceae, SHA_31,* *Erysipelotrichaceae,* *Bradyrhizobiaceae, Hyphomicrobiaceae,* *mitochondria,* *Alcaligenaceae,* *Comamonadaceae, Myxococcaceae,* *Enterobacteriaceae,* *auto67_4W* *Genera:* *Geothrix, Bifdobacterium, Paenisporosarcina,* *Alloiococcus, Caloramator,* *Allobaculum, Rhodoplanes,* *Carludovica,* *Novosphingobium, Dechloromonas, Klebsiella,* *Coxiella,* *Pseudomonas*
					Differentially abundant taxa between normal and RCC groups: *Pseudomonas* *Klebsiella* *Carludovica* *Phyllobacterium* *Rhodoplanes* *Allobaculum* *Chloroplast* *Streptophyta* *Rickettsiales* *Deinococcus*	
Differences in the Microbiome between Responders and Non-Responders to ICI Therapy						
Study	Year of Publication	Number of RCC Patients	Method	Microbiome Diversity in ICI Responders	Specific Microbiome in Responders	Specific Microbiome in Non-Responders
Agarwal et al. [50]	*2020*	22	16s rRNA sequencing	No significant difference	*Akkermansia muciniphila* *Verrucomicrobiae bacteria*	*Unspecified*
Derosa et al. [61]	*2020*	67	Whole genome sequencing	Increased	** *Akkermansia muciniphila* ** *Bacteroides salyersiae* *Eubacterium siraeum*	*Clostridium Clostridioforme* *Clostridium hathewayi* *Erysipelotrichaceae bacterium*
Routy et al. [47]	*2018*	40	Shotgun metagenomic sequencing	Increased	** *Akkermansia muciniphila* ** *Alistipes* sp. *Eubacterium* sp. *Furmicutes* sp. *Intestinihomonas* *Ruminococcaceae* sp.	*Bacteroides nordii* *Parabacteroides Distasonnis* *Proteobacteria*
Salgia et al. [48]	*2020*	31	Shotgun metagenomic sequencing	Increased	** *Akkermansia muciniphila* ** *Bacteroides eggerthii* *Barnesiella intestine hominis* *Bifidobacterium adolescentis* *Faecalibacterium* sp. *Firmicutes bacterium* *Odoribacter splanchnicus* *Prevotella copri* *Prevotella* sp. *Ruminococcus torques*	*Bacteroides ovatus* *Eggerthelia lenta* *Flavonifractor plautii* *Fusicatenibacter* *saccharivorans*
Meza et al. [58]	*2022*	28	RNA sequencing	Unspecified	*Cutibacterium acne*, *Moraxella osloensis* and *Pasteurella multocida* were abundant in both responders and non-responders.Increased in responders: *Stenotrophomonas maltophilia* *Corynebacterium* sp.	
Combination of Microbiome with ICI Therapy						
Study	Year of Publication	Number of RCC Patients	Method	Management	Efficacy Outcome	Microbiome Analysis
Dizman et al. [59]	*2022*	29	Stool metagenomic sequencing	Ipilimumab (anti-CTLA4) and nivolumab (anti-PD1) + live bacterial product CBM588 contains *Clostridium butyricum*	Patients treated with “nivolumab–ipilimumab plus CBM588” showed better PFS and OS than those treated with “nivolumab–ipilimumab”. No significant difference in toxicity was observed between the two groups.	In patients receiving CBM588: *Bifidobacterium* sp., *Bifidobacterium longum* and *Butyricimonas faecalis* increased in responders. *Desulfovibrio* sp. decreased in responders
Clinical Outcomes of Concomitant Use of Factors that Alter the Gut Microbiome (Antibiotics, PPI) with ICIs						
Study	Year of Publication	Number of RCC Patients	Treatment	Effect of Treatment		
Giordan et al. [49]	*2021*	33	Antibiotics/PPI+ post-ICI(anti-PD1)	Use of antibiotics and PPIs alone or combined negatively associated with PFS and OS.		
Mollica et al. [50]	*2022*	62 (Cohort 1); 156 (Cohort 2)	Pre-ICI: ipilimumab (anti-CTLA4) and nivolumab (anti-PD1) + PPI (Cohort 1); Pre-ICI: nivolumab (anti-PD1) + PPI (Cohort 2)	Concomitant use of PPI with ICIs did not affect survival outcomes.		
Tsikala-Vafea et al. [49]	*2021*	*603:* *121* [61] *12* [62] *35* [63] *55* [64] *146* [65] *29* [66] *67* [47] *25* [67] *65* [68] *48* [69]	Antibiotics + post-ICI	The use of antibiotics was associated with shorter PFS and shorter OS.		
Hahn et al. [74]	*2018*	*145*	Pre-VEGF-TKI + antibiotics targeting Bacteroides species	Targeting stool *Bacteroides* sp. with antibiotics improves PFS in patients receiving first-line VEGF-TKIs in a duration-dependent manner.		
The Microbiome as a Prognostic Biomarker						
Study	Year of Publication	Number of Specimens	Discoveries			
Kovaleva et al. [45]	*2022*	Seventy-seven FFPE tissue samples (23 ccRCC, 19 pRCC and 24 chRCC; 11 normal kidney tissues)	A negative correlation identified between bacterial burden and the stromal inflammatory markers in kidney tumors. In ccRCC patients, higher bacterial burden and increased number of stromal cells were associated with a poor prognosis.			

RCC, renal cell carcinoma; ccRCC, clear cell RCC; pRCC, papillary RCC; chRCC, chromophobe RCC; FFPE, formalin-fixed paraffin-embedded; VEGF, vascular endothelial growth factor; TKI, tyrosine kinase inhibitor; ICI, immune-checkpoint inhibitor; PPI, proton-pump inhibitor; PFS, progression-free survival; OS, overall survival.

## Data Availability

The data presented in this study are available from the corresponding author upon reasonable request.

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
