# Peer review of "A New Treatment Landscape for RCC: Association of the Human Microbiome with Improved Outcomes in RCC"

_cancers, 2023, doi:10.3390/cancers15030935_

Round 1
Reviewer 1 Report
It was my pleasure to review the article: A New Treatment Landscape for RCC: Association of the Human Microbiome with Improved Outcomes in RCC by Seok Joong Yun et al.
The review are interesting to the development of treatment for RCC and the manuscript is well written. I only have a minor point to be addressed:
1) Line 309,310.
A systematic review for the effect of antibiotics on immune-checkpoint blockade was reported by Wilson. B. E (https://doi.org/10.1007/s00262-019-02453-2). Please add to your article.
Author Response
Thank you for the reviewer's comment.
The reference has been added (ref. 70).
Reviewer 2 Report
Dear authors: This review is an excellent easy-to-read educational review to the readers. All the languages are comprehensive and all the sections are well-organized. The information collected in the paper seem to include all the current RCC treatments associated with microbiome. This is a good review.
I would like to suggest a few points here:
1. The review paper would need a clear figure abstract to demonstrate the the roles of microbiome in RCC. This would make the reading more straightforward.
2. There are two '3.1' and three '4.1's in section 3 or 4. Please correct them.
3. In the tables, there are some duplicated tables. For example, there are two 'The microbiome as a prognostic biomarker'. There are more than one duplicates, please look into them carefully and keep only one for the paper.
4. In the section 4, the data suggested that antibiotic could have no impact or negative impact on survival rate of patient while some probiotics could help extend the life expectancy. Is there any molecular biology analysis in any of these experiments? I hope the author could at least analyze the mechanism found between microbiome and ICIs. For example, is there any cell signaling/biomarkers that the microbiome implantation such as CBM588 could activate the leukocytes? Is there any protein level varied after using antibiotics so that the activation of immune response got lower? If there is no detailed evidence from the previous publication, there is a room for author to discuss these further and make some future experiment suggestions.
5. In the conclusion, the author only summarized the current data from clinical treatments and on-going trials. There would be a need for future direction suggestion in this section. For example, some targeted antibiotic treatment like reference 74 is to indicate a great future direction of RCC treatment, and a possible Nanoparticle delivery technology can be applied for microbiome intervention (https://www.nature.com/articles/s41565-019-0589-5).
